# 21 Fluorescent Protein-Based DNA Staining Dyes

**DOI:** 10.3390/molecules27165248

**Published:** 2022-08-17

**Authors:** Yurie Tehee Kim, Hyesoo Oh, Myung Jun Seo, Dong Hyeun Lee, Jieun Shin, Serang Bong, Sujeong Heo, Natalia Diyah Hapsari, Kyubong Jo

**Affiliations:** 1Department of Chemistry, Sogang University, 35 Baekbeom-ro, Mapogu, Seoul 04107, Korea; 2Chemistry Education Program, Department of Mathematics and Science Education, Sanata Dharma University, Yogyakarta 55282, Indonesia

**Keywords:** DNA, single-molecule, microfluidic device, fluorescent protein, DNA-binding proteins, FP-DBP

## Abstract

Fluorescent protein–DNA-binding peptides or proteins (FP-DBP) are a powerful means to stain and visualize large DNA molecules on a fluorescence microscope. Here, we constructed 21 kinds of FP-DBPs using various colors of fluorescent proteins and two DNA-binding motifs. From the database of fluorescent proteins (FPbase.org), we chose bright FPs, such as RRvT, tdTomato, mNeonGreen, mClover3, YPet, and mScarlet, which are four to eight times brighter than original wild-type GFP. Additionally, we chose other FPs, such as mOrange2, Emerald, mTurquoise2, mStrawberry, and mCherry, for variations in emitting wavelengths. For DNA-binding motifs, we used HMG (high mobility group) as an 11-mer peptide or a 36 kDa tTALE (truncated transcription activator-like effector). Using 21 FP-DBPs, we attempted to stain DNA molecules and then analyzed fluorescence intensities. Most FP-DBPs successfully visualized DNA molecules. Even with the same DNA-binding motif, the order of FP and DBP affected DNA staining in terms of brightness and DNA stretching. The DNA staining pattern by FP-DBPs was also affected by the FP types. The data from 21 FP-DBPs provided a guideline to develop novel DNA-binding fluorescent proteins.

## 1. Introduction

Fluorescent proteins (FP) have been essential molecular reporters for microscopic visualization in biochemical and cellular applications since the first application in 1994 [1]. FPs have been applied for monitoring gene expression, protein localization, protein dynamics, and protein-DNA interactions from the molecular to the cellular level [2]. FPs have distinct advantages as they are easy to use and inexpensive. Most FP genes can be obtained through a non-profit plasmid repository, AddGene [3]. Using genetic engineering tools, an FP gene can be placed at either the N- or C-termini of a gene of interest [4]. Then, the expression vector can be transported into a host cell with an organelle-targeting signal sequence which allows the expression of FPs in vivo. Moreover, FP genes can be readily mutated to improve various features, such as excitation and emission wavelengths, brightness, pKa, maturation, lifetime, and photostability. Since the first success of GFP mutation in 1994 [5], many new and improved FPs have been genetically engineered. To date, the FP database (FPbase.org) lists around 800 FPs, and the numbers are still increasing [6]. Each newly developed mutant FP has improved features. For example, the emission wavelengths have been expanded from a green color of 509 nm [7] to a spectrum of wavelengths from 424 nm [8] to 1000 nm [9]. As described so far, FPs have been developed to have many advantages, but they also have disadvantages. A primary limitation of FP is low stability, especially photostability, attributed to protein’s properties such as denaturation or degradation. Therefore, there have been efforts to solve this problem through genetic engineering. Recently, a highly stable green fluorescent protein, StayGold, was reported to overcome the durability limitation [10].

Brightness is the most important property of FP as it determines the detection sensitivity and the linear dynamic range for quantitative assays. The brightness is defined as the product of an extinction coefficient and a quantum yield (*b* = *ε* × *ϕ*) [11]. The first wild-type *A*. *victoria* GFP has a relatively low brightness (*b* = 19.75 mM^−1^cm^−1^) [12]. Genetic engineering improves FP brightness markedly higher, even compared with the brightest organic dye commercially available [13,14]. Currently, AausFP1 is the brightest FP, *b* = 164.9 mM^−1^cm^−1^, eight-fold brighter than the original wild-type GFP, and nearly five-fold brighter than popular EGFP (*b* = 33.54) [15]. Among red FPs, RRvT is the brightest FP (*b* = 117.92), seven-fold brighter than popular mCherry (*b* = 15.84) [16].

Fluorescent proteins can be genetically fused with DNA-binding peptides or proteins (FP-DBP) to visualize DNA molecules [2,17]. So far, various methods have been developed to obtain sequence information from elongated DNA molecules. The optical mapping is a representative platform [18,19,20] and is now commercially available via BioNano Genomics [21]. Most methods have been based on the use of intercalating fluorescent dyes such as YOYO-1, A/T-specific DNA-binding chemicals [22,23,24], and fluorochrome addition to nucleotides [25,26,27]. However, organic dyes have intrinsic limitations. Therefore, FP-DBPs have the potential to replace most single-molecule DNA applications using fluorescent organic dyes. For example, Kang et al. applied FP-DBPs for a DNA curtain experiment [28]. Their binding was reversible by adjusting buffers, and they do not cause photo-induced DNA breaks because their fluorophore is buried in their barrel. A notable advantage of using FP-DBP is that the number of DNA-binding peptides or proteins is enormous. Most of them are potential candidates for staining or labeling DNA molecules. Furthermore, FP-DBP can be connected to nanoparticles, which can allow the DNA molecules to become observable under an electron microscope [29,30]. FP-DBP has been also utilized in a cell for visualizing DNA and its associated biomolecules [31]. FP-DBPs can readily be loaded into the cell nucleus through general molecular biology techniques [32].

Previously, we developed FP-DBPs as DNA-staining reagents for single-molecule visualization [17]. Our first FP-DBP consisted of eGFP or mCherry linked with small DNA-binding peptides at both ends (KWKWKKA-FP-AKKWKWK). Next, we reported various DNA-binding peptides, such as lysine heptamers, SPRK repeats, and high mobility group (HMG: TPKRPRGRPKK) [33]. We further expanded on the histone-like nucleoid-structural protein (H-NS), high mobility group (HMG) [34], and truncated transcription activator-like effector (tTALE) [35] to show the AT-specific binding of FP-DBPs. Since FP-DBPs can be designed to stain DNA with sequence-specific patterns, FP-DBP-generated specific patterns can be quite useful for identifying the directions and positions in the genomic map.

In this paper, we recombined HMG and tTALE with various FPs as an expansion from eGFP and mCherry. Through the recombination, ten FPs were connected to HMG, and eleven FPs were connected to tTALE. In total, 21 FP-DBPs were successful in staining DNA. These fluorescent proteins have different colors, and different brightness compared with reference data. We measured the fluorescent intensity of flow-driven elongated DNA molecules tethered to a biotinylated surface. In addition, the importance of FP to DBP connecting orientation was revealed by RRvT and mNeonGreen with HMG. Using tTALE-FPs, we also demonstrated that A/T-specific staining was also possible with other FPs. However, their characteristics were different depending on the kind of FPs. The data from 21 FP-DBPs provided a guideline to develop novel DNA-binding fluorescent proteins.

## 2. Results and Discussion

### 2.1. Construction and Expression of Diverse HMG-FPs and tTALE-FPs

Figure 1 shows our recombined 21 FP-DBPs and their stained λ DNA (48.5 kb). In our previous studies, HMG and tTALE were linked only with eGFP and mCherry [34,35]. In order to enhance the brightness and expand emission wavelengths, we constructed 21 plasmids by connecting 14 FP genes obtained from AddGene. In our previous study, we linked HMG peptides at both N- and C-termini of an FP [34], but in this study, we connected HMG at the N-terminus of FP for the efficiency of the cloning processes. N-terminal linkage was successful in most cases as shown in Figure 1a. When HMG-FP could not be expressed, HMG was recombined into the C-terminus of FP. For example, when HMG was linked at the N-terminus (HMG-tdTomato), the color of tdTomato was not obtained. However, when HMG was linked at the C-terminus (tdTomato-HMG), the expression was successful. Since tTALE is 36 kDa, it was always linked to the N-terminus of eleven FPs. After forming the constructs, we screened their expression with BL21 E. coli. Since tTALE has a complex structure, expressions of FP linked to tTALE were generally challenging, particularly when the FP oligomerization state was not monomer (Table 1).

Table 1 shows 14 kinds of FPs that we used for FP-DBPs and three references, such as AasuFP1, eGFP, and avGFP. We selected bright FPs as much as possible. Since AausFP1 is the brightest, we also attempted to construct HMG-AausFP1. Unfortunately, HMG-AausFP1 had a high background, which made it difficult to visualize DNA backbones. Further, HMG-AausFP1 aggregated on the DNA backbone to form bright spots instead of linear lines. However, we confirmed bright AausFP1 that was fused with a different protein. For tTALE, monomeric FPs were more likely expressed, probably because tTALE was large. This fact limited expressions with AausFP1, RRvT, and tdTomato which are supposed to be brighter than monomeric FPs.

### 2.2. Brightness Optimization of FP-DBPs

The brightness of an FP is critical for a single-molecule observation. As brightness becomes higher for an FP, a lower amount of the FP is required for detection. We referred to a fluorescent protein database (FPbase.org) for the selection. However, the database does not account for how brightness would change when an FP is coupled to a DBP. Further, the microscopic set-up with emission and excitation filter affects the measured intensity of the brightness of FP. In addition, the number of FP-DBPs per unit length of DNA also affects the fluorescence intensity measured on the image. Given these concerns, we measured the fluorescence intensity of HMG-FPs and tTALE-FPs and compared them to the brightness shared by the FP database (Figure 2). To measure their brightness from the microscopic images, we selected 60 different DNA molecules and measured the brightest fluorescence intensity within a DNA molecule with ImageJ software.

Figure 2 reveals the comparison between FP brightness and the fluorescence intensities of FP-DBP-stained DNA images. The order of fluorescence intensity roughly matched the FP database brightness. For example, the fluorescence intensity for mCherry was darkest for both whereas RRvT linked with HMG (Figure 2a) and mNeonGreen with tTALE (Figure 2b) were brightest. The dashed line calculated for a linear relationship between FP brightness and FP-DBP fluorescence intensity can be a general guideline. FPs linked to HMG correlated better than FPs linked to tTALE. Because tTALE is larger than HMG, tTALE is more prone to affect the brightness of an FP. Additionally, the correlation of tTALE constructs reflected that. Surprisingly, a few exceptions had brighter fluorescence intensity than what was referred to by the FP database such as HMG-Emerald, tTALE-mTurquoise2, and tTALE-mKO2. The brightness jump of tTALE-mTurquoise2 is more prominent when compared to HMG-mTurquoise2 (Figure 2b). A weak dimer, HMG-Ypet, and a tandem dimer, tdTomato-HMG, were dimmer. However, tTALE-Ypet was one of the brightest tTALE-FPs. Therefore, we concluded that generally, the intensity of FP-DBP can depend on the FP brightness, but there are many exceptions. A plausible explanation can be that the brightness of the FP was affected by protein folding. An FP that is linked after a protein motif can sometimes be affected by how well the protein motif folds, and we suspect that tTALE-mTurquoise2 might have received a beneficial effect during the process. To explain this observation, we measured fluorescence spectra using mNeonGreen-conjugated FP-DBPs. Figure 2c demonstrates two important factors to change brightness. First, two different DBPs affect the fluorescence intensity and spectrum shape. Second, DNA-bound FP-DBPs have enhanced and slightly different fluorescence spectra.

Figure 3 demonstrates the positional effects between FP and DBP. RRvT-HMG was 10% brighter, though they were in the error range, than HMG-RRvT, even though amino acid compositions were essentially the same (Figure 3a). Figure 3b compares the effect of FP position on DNA-staining behavior. For HMG-mNeonGreen, DNA was aggregated instead of stretching on a positively charged surface. mNeonGreen-HMG, on the other hand, allowed DNA to be well stretched. This observation explains that the placement of an FP and a DBP can affect the brightness of FPs.

### 2.3. A/T-Rich Specific Staining by tTALE-FP

One of the important advantages of FP-DBP as a dye is its affinity toward a specific DNA sequence. Nonetheless, selective binding exclusively toward a specific sequence on a single molecular level has been a challenging task [37]. In addition to the optical mapping system described in the introduction, fluorescence in situ hybridization (FISH) has been a tool to obtain sequence information from metaphase chromosomes. Further, DNA points accumulation for imaging in nanoscale topography (DNA-PAINT) can enhance the resolution of DNA images [38,39]. However, the hybridization method is based on DNA melting to expose single-stranded DNA. In contrast, DNA-binding proteins can recognize the sequence without opening the double-helix. Thus, we chose protein to direct specific binding; transcription activator-like effector (TALE) and Zinc-Finger domain (ZnF), for instance, are known for their specific sequence affinity. Yet, they are also known for their false positive binding [40]. In our previous report, we showed the A/T-specific binding of tTALE-FP by increasing salt concentrations [35]. The results led us to question, “would the A/T specificity be affected by an FP that is linked by the DBP?” In testing so, we utilized five tTALE-FPs to an A/T-specific stain λ phage DNA with tTALE-Emerald, mStrawberry, Ypet, mNeonGreen, and mTurquoise2. Since tTALE with Emerald and mCherry bound the DNA with A/T specificity [35], Emerald was used as a control for our experiments in this paper.

Figure 4 demonstrates DNA-staining with tTALE-FPs by increasing salt concentrations. Five tTALE-FPs were Emerald, mTurquoise2, Ypet, mNeongreen, and mStrawberry. The DNA images of Emerald showed standard A/T-specific staining of DNA. As the salt concentration increased, the 5′-end GC-rich region of the λ phage DNA was broadly destained. Other tTALE-FPs also stained the DNA with distinct patterns. tTALE-mTurquoise2, tTALE-Ypet, and tTALE-mStrawberry stained with patterns resembling tTALE-Emerald. Interestingly, the salt concentrations at which the A/T rich regions appeared distinctively were different for different FPs. From the context, tTALE-mStrawberry caught our attention for the A/T-specific pattern did not appear until 100 mM of salt was applied. The strong interaction strength helps us to develop a DNA staining dye working in a relatively high salt condition. tTALE-mNeonGreen did not generate a typical tTALE-FP pattern. Instead, tTALE-mNeonGreen’s pattern appeared to be related to TGTCTGT patterns that truncated TALE were supposed to bind based on TALE binding rule.

To explain the differences in the DNA staining patterns of different tTALE-FPs, we compared protein sequences and structures. Figure 5a shows a phylogenetic tree generated by MEGA (molecular evolutionary genetic analysis) software. Emerald, mTurquoise2, and YPet are closely related. Figure 5b shows the overlapped structure of three FPs, which are almost identical to one another. In contrast, mStrawberry, and mNeonGreen were significantly apart from the three FPs. Figure 5c shows the overlapped structure of Emerald, mStrawberry, and mNeonGreen. They show differences among them. These differences may explain the different profiles of DNA-bound tTALE-FPs in Figure 4.

When we traced the FP engineered history, Emerald, mTurquoise2, and Ypet shared their originality to avGFP; mNeongreen was a derivative of LanYFP; mStrawberry was a derivative of DsRed. This similarity of avGFP-derivatives reflected their similarity in the salt concentration at which the A/T-specific pattern appeared. The FP sequence deviation would alter how FPs fold after tTALE. If protein sequences are similar to each other, effects from the protein folding would be most likely similar.

In this paper, we developed and characterized 21 FP-DBPs to increase brightness and optimize properties. The 21 FP-DBPs were used to stain DNA molecules and analyze fluorescence intensities. HMG was used to demonstrate the importance of the binding sequence between FP and DBP. Using tTALE-FP, we found different specificities of tTALE depending on FP. Understanding this interaction between FP and DBP can be a stepping stone towards the engineering and optimization of protein-based DNA staining dyes. This discovery will also enable genetic engineering to expand DNA staining dyes toward multi-color staining.

## 3. Materials and Methods

### 3.1. Chemicals

DNA primers and oligonucleotides were purchased from COSMOGENETH (Seoul, Korea). Biotin-labeled DNA oligomer, 1 kb, and 100 bp ladder were purchased from Bioneer (Daejeon, Korea). E. coli strain DH5α and BL21 (DE3) were from Yeastern (Taipei, Taiwan). pET-15b plasmids were from Novagen, (Darmstadt, Germany). Bacteriophage λ DNA (48.5 kb) was purchased from New England Biolabs (NEB). Ni-NTA agarose resin and disposable empty gravity column were from Qiagen. N-Trimethoxymethyl silyl propyl-N,N,N-trimethyl ammonium chloride in 50% methanol was purchased from Gelest. Epoxy was from Permatex (Solon, OH, USA). Other enzymes were purchased from NEB (Ipswich, MA, USA), and other chemicals were from Sigma-Aldrich (St. Louis, MI, USA).

### 3.2. FP-DBP Recombinants

The plasmids were constructed for HMG-fused fluorescent proteins and tTALE-fused-fluorescent proteins. Each HMG-FP plasmid was created by an extension polymerase chain reaction which links fluorescent protein to the N- or C-terminal of the DNA-binding protein, HMG. The amino acid sequence of a linker between the DNA-binding protein and the fluorescent proteins was GGSGG (5′-GGA GGC TCG GGC GGG-3′). For the construction of HMG-FPs, constructs were requested from Addgene. For mCherry, pLV-mCherry was a gift from Pantelis Tsoulfas (Addgene plasmid # 36084). For RRvT [16], pBad-HisB-RRvT was a gift from Robert Campbell (Addgene plasmid # 87364). For tdTomato [41], pCAG-tdTomato was a gift from Angelique Bordey (Addgene plasmid # 83029). For mOrange2 [42], mOrange2-C1 was a gift from Michael Davidson and Roger Tsien (Addgene plasmid # 54650). For mStrawberry [43], mStrawberry-N1 was a gift from Michael Davidson, Nathan Shaner and Roger Tsien (Addgene plasmid # 54644). For YPet [44], YPet-N1 was a gift from Patrick Daugherty and Michael Davidson (Addgene plasmid # 54637). For mClover3, mClover-pBAD was a gift from Michael Davidson (Addgene plasmid # 54805). For mNeonGreen [45], Lamp1-mNeonGreen was a gift from Dorus Gadella (Addgene plasmid # 98882). For Emerald, Emerald-N1 was a gift from Michael Davidson (Addgene plasmid # 54588). For mTurquoise2 [46], mTurquoise2-N1 was a gift from Michael Davidson and Dorus Gadella (Addgene plasmid # 54843). The Addgene constructs served as a template to form an extension PCR product that incorporates HMG and GGSGG linker. The two sets of forward primer and the reverse primers for N- and C terminal fusion of HMG were used. For N-terminal tagging, the forward primer (5′-ATG TTG CAT ATG ACT CCC AAG CGT CCC CGC GGG CGC CCC AAG AAG GGA GGC TCG GGC G GG ATG GTG AGC AAG GGC GAG G-3′) and the reverse primer (5′-ATG TTG GGA TCC TTA CTT GTA CAG CTC GTC CAT G-3′) and for C terminal tagging, (5′-ATG TTG CAT ATG ATG GTG AGC AAG GGC GAG GAG G-3′) and the reverse primer (5′-ATG TTG GGA TCC TTA TGT CGG CTT ACG CGG GCG CCC ACG AGG TTT TTT CCC GCC CGA GCC TCC CTT GTA CAG CTC GTC CAT G-3′) TALEN plasmid was isolated from RHD_Exon4_TALEN_L [35]. tTALE gene was first introduced into the pET- 15b vector with two restriction sites, NdeI and XmaI sites. For adding restriction sites to the tTALE sequence, forward primer (5′-ATG TTG CAT ATG GAT CTA CGC ACG CTC GGC TAC -3′) and reverse primer (5′-ATG TTG GGA TCC ATG TTG CCC GGG GCC GCC AGA GCC GCC CCC ATG ATC CTG ACA CAA AAC AGG CAA C-3′) were used in the PCR process. For FP introduction to tTALE-FPs, four Addgene constructs were requested additionally. For mScarlet [47], pEB2-mScarlet was a gift from Philippe Cluzel (Addgene plasmid # 104006). For mKO2 [48], mKO2-N1 was a gift from Michael Davidson and Atsushi Miyawaki (Addgene plasmid # 54625). For mVenus [49], mVenus N1 was a gift from Steven Vogel (Addgene plasmid # 27793). For mEos4b [50], pRSETa_mEos4b was a gift from Loren Looger (Addgene plasmid # 51073). Fluorescent protein (mScarlet, mKO2, mOrange2, mStrawberry, mVenus, mClover3, mNeonGreen, mEos4b, Emerald, and mTurquoise2) plasmids were used as templates to create PCR Products that fused with GGSGG linker and XmaI restriction site. tTALE- mCherry and tTALE-YPet were constructed by overlapping extension PCR method. The fluorescent protein gene and tTALE gene were amplified and fused with a linker sequence (5′-CTT GTA CAG CTC GTC CAT GCC-3′) by the extension PCR process. For the FP gene, a forward primer (5′-GGC GGC TCT GGC GGC ATG GTG AGC AAG GGC GAG G-3′) and a reverse primer (5′-ATG TTG GGA TCC TTA CTT ATA GAG CTC GTT CAT GCC CTC GG-3′) were used in 50μL of Pfu PCR PreMix (Bioneer), for tTALE gene, a forward primer (5′-ATG TTG CAT ATG GAT CTA CGC ACG CTC GGC TAC-3′) and a reverse primer (5′-GCT CTT CGC CTT TGC TCA CCA TGC CGC CAG AGC CGC CCC CAT GAT CCT GAC ACA AAA CAG GC-3′) were used for the PCR process. For the final extension step, after both the tTALE and FP inserts were linked, a forward primer (5′-ATG TTG CAT ATG GAT CTA CGC ACG C-3′) and a reverse primer (5′-ATG TTG GGA TCC TTA CTT ATA GAG CTC GT ATG TTG GGA TCC TTA CTT ATA GAG CTC GT-3′) were used.

### 3.3. Molecular Cloning

Using standard subcloning procedures, HMG-FP sequences were inserted into the pET-15b vector and transformed into the E. coli BL21 (DE3) strains by using NdeI and BamHI restriction. For the tTALE-FPs, FP sequences were inserted into the same vector and transformed by XmaI and BamHI digestion instead of NdeI. A single colony of the transformed cells was inoculated in a fresh LB media containing ampicillin and incubated for 1 h. After the transformed cells were saturated, they were subsequently cultured to an optical density of ~0.4 at 37 °C with corresponding antibiotics. The over-expressed and FP-tagged proteins were induced with a final concentration of 1 mM for IPTG overnight on a shaker at 20 °C and 150 rpm. The cells for the protein purification were harvested by centrifugation at 10,000× *g*, for 10 min (following centrifugations were performed under similar conditions), and the residual media was washed using the cell lysis buffer (50 mM Na_2_HPO_4_, 300 mM NaCl, 10 mM Imidazole, pH 8.0). The cells were lysed by ultrasonication for 30 min, and the cell debris was centrifuged at 13,000 rpm for 10 min at 4 °C. The his-tagged FP-DNA-binding proteins were purified using affinity chromatography with Ni-NTA agarose resin. The mixture of crude extract and the resin were kept on a shaking platform at 4 °C for 1.5 h. The lysate containing proteins bound Ni-NTA agarose resin was loaded onto the column for gravity chromatography and was further rinsed several times using the Nickel-NTA wash buffer (50 mM Na_2_HPO_4_, 300 mM NaCl, 20 mM Imidazole, pH 8.0) several times. Finally, the bound proteins were eluted using Elution buffer (50 mM Na_2_HPO_4_, 300 mM NaCl, 250 mM imidazole, pH 8.0). All proteins were diluted (10 μg mL^−1^) using 50% *w*/*w* glycerol/1× TE buffer (Tris 10 mM, EDTA 1 mM, pH 8.0).

### 3.4. Polydimethylsiloxane (PDMS) Microfluidic Devices

A standard rapid prototyping method was used to create PDMS microfluidic devices for DNA elongation and deposition on a positively charged surface [51]. Briefly, the patterns on a silicon wafer for microchannels (2.3 μm high and 100 μm wide) were fabricated using SU-8 2005 photoresist (Microchem, Netonpression, MA, USA). The PDMS pre-polymer mixed with a curing agent (10:1 weight ratio) was cast on the patterned wafer and cured at 65 °C for four hours or longer. Cured PDMS was peeled off from the patterned wafer, and the PDMS devices were treated in an air plasma generator for 1 min with 100 W (Femto Science Cute Basic, Hwaseong, Korea) to alter the PDMS surface to become hydrophilic. The PDMS devices were punctured for an inlet and outlet. The devices were stored in water before use.

### 3.5. Positively-Charged Surface Preparation

Glass coverslips were stacked in the Teflon rack, soaked in a piranha etching solution (30:70 *v*/*v* H_2_O_2_/H_2_SO_4_) for 3 h, and rinsed with deionized water until the pH reached the neutral pH (pH 7). For the glass surfaces, 350 μL of N-trimethoxymethylsilylpropyl-N,N,N-trimethyl ammonium chloride in 50% methanol was mixed with 200 mL of deionized water. The acid-cleaned glass coverslips were incubated in this solution for 12 h at 65 °C. Then, they were rinsed with ethanol three times. The surfaces were stored in ethanol before use.

### 3.6. Microscopy

The microscopy system consisted of an inverted microscope (Olympus IX70, Tokyo, Japan) equipped with a 100× Olympus UPlanSApo oil immersion objective lens and illuminated LED light source (SOLA SM II light engine, Lumencor, Beaverton, OR, USA). The light was passed through the corresponding filter sets (Table 2) to excite the fluorescent dye. Fluorescence images were captured using a scientific-grade complementary metal-oxide-semiconductor digital camera (2048 × 2048, Prime sCMOS Camera, Photometrics, Tucson, AZ, USA) and stored in 16-bit TIFF format generated by the software Micro-manager. ImageJ was utilized for image processing, particularly to stitch the microchannel images (Figure 2b).

### 3.7. DNA Imaging on a Positively Charged Surface and Fluorescence Intensity Measurement

DNA molecules (15 ng µL^−1^) were mixed 1:1 with FP-DBPs (5–100 nM) in 1× TE (10 mM Tris, 1 mM EDTA, pH 8). The DNA and FP-DBPs were incubated at room temperature for 10 min. The mix was loaded on a positively-charged surface and was allowed to spread between the surface and a glass slide. From the fluorescence images, we randomly selected 60 DNA molecules and measured the brightest fluorescent intensity within a DNA molecule. The measurement was performed with ImageJ software after background subtraction. In total, 60 values were combined and averaged for a fluorescence intensity value.

### 3.8. DNA Imaging on a Positively Charged Surface with Microchannels

DNA molecules (60 ng µL^−1^) were mixed 1:1 with FP-DBPs (20–70 nM) in 1× TE (10 mM Tris, 1 mM EDTA, pH 8) with NaCl. The NaCl concentration was twice the targeting concentration before mixing. The mix was incubated on ice for 10 min. The mix was then diluted at about 1/200 with 1× TE with a targeting concentration of NaCl. The PDMS device was washed with ethanol and water. When the device was dried, it was placed on a positively-charged surface. The diluted mix was loaded through the inlet hole of the device.

### 3.9. Fluorescence Intensity Measurement with the Fluorometer

In total, 1 μM of FP-DBP was either mixed or not mixed with 250 pM (7.88 ng µL^−1^) of λ Phage DNA in 10 mM phosphate pH 8.0 buffer. When the DNA samples were mixed with DNA, the samples were incubated on ice for 10 min before measurements. Using a Hitachi F-7000 fluorometer, emission spectrums were collected using the fluorescent protein’s maximum excitation wavelength.

## Figures and Tables

**Figure 1 molecules-27-05248-f001:**
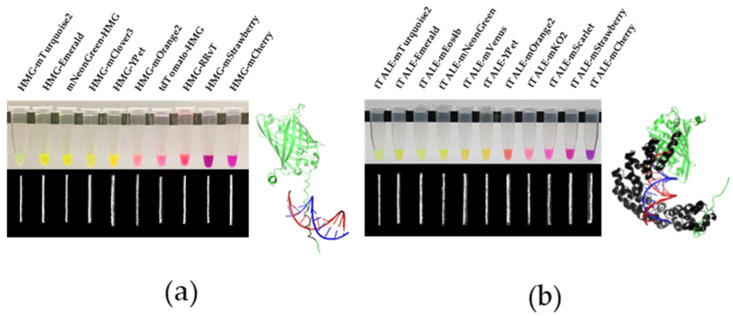
**Expressed FP-DBPs in test tubes and their stained DNA images.** (**a**) HMG-FP/FP-HMG (**b**) tTALE-FP. FP-DBP stained λ phage DNA images were given below. Illustrations of FP-DBP binding to DNA were shown for comparison of the size of DBPs. HMG-Emerald and tTALE-Emerald structures were modeled through AlphaFold 2.1.0 (DeepMind, London, UK) [36]. DNA-binding protein motifs were colored black and were aligned to DNA by using PDB 2EZD (**a**) PDB 4OTO (**b**) in the software PyMOL.

**Figure 2 molecules-27-05248-f002:**
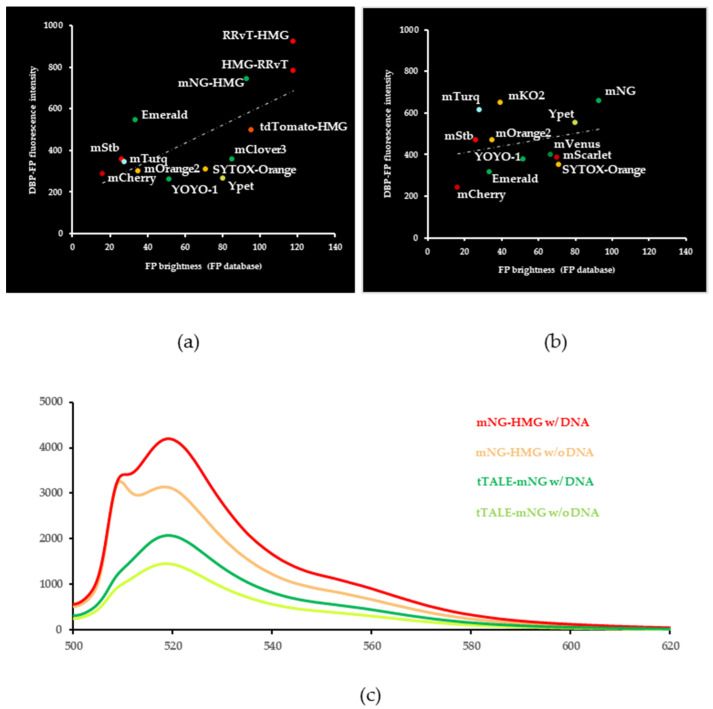
**Fluorescence comparison between reference and measured values.** (**a**) HMG (**b**) tTALE linked FP brightness was obtained from the FPbase.org (*X*-axis), and FP-DBP fluorescence intensities were measured from microscopic images (*Y*-axis). If the DBP name was omitted, DBP was linked N-terminal to FP. The dotted lines are calculated for a linear relationship. R^2^ was 0.48 for HMG constructs (**a**) and 0.08 for tTALE constructs (**b**). FP abbreviated names: mNG, mNeonGreen; mStb, mStrawberry; mTurq, mTurquoise2. (**c**) Fluorescence spectra of mNG-HMG and tTALE-mNG with and without DNA.

**Figure 3 molecules-27-05248-f003:**
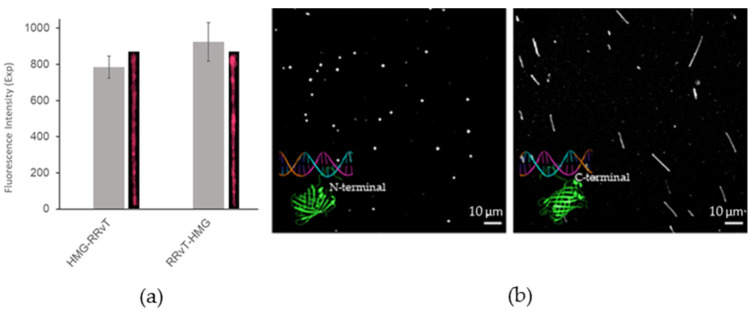
**Positional effects of FP and DBP****.** (**a**) Integrated intensity comparison of HMG-RRvT vs. RRvT-HMG. The λ DNA molecules were stained with FP-DBP. (**b**) Stretching comparison of HMG-mNeonGreen (N-terminal) vs. mNeonGreen-HMG (C-terminal). Scale bars = 10 μm.

**Figure 4 molecules-27-05248-f004:**
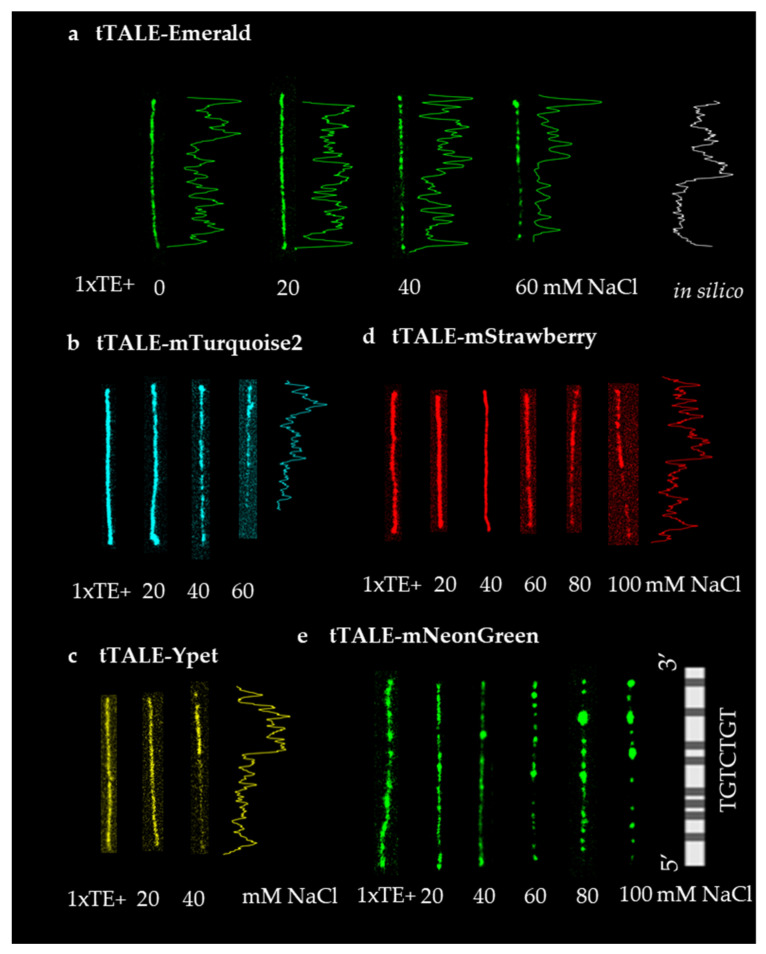
**tTALE-FP staining of λ Phage DNA for A/T specific map.** (**a**) λ Phage DNA was stained with 30 nM tTALE-Emerald. The fluorescence intensity profiles are shown along with the in silico profile of AT frequency. Stained images of (**b**) 40 nM tTALE-mTurquoise2 (**c**) 40 nM tTALE-Ypet (**d**) 30 nM tTALE-mStrawberry and an intensity profile were shown. A stained image of (**e**) 30 nM tTALE-mNeonGreen with in silico maps of TGTCTGT was shown. Fully stained DNA was treated with different concentrations of NaCl in 1× TE buffer.

**Figure 5 molecules-27-05248-f005:**
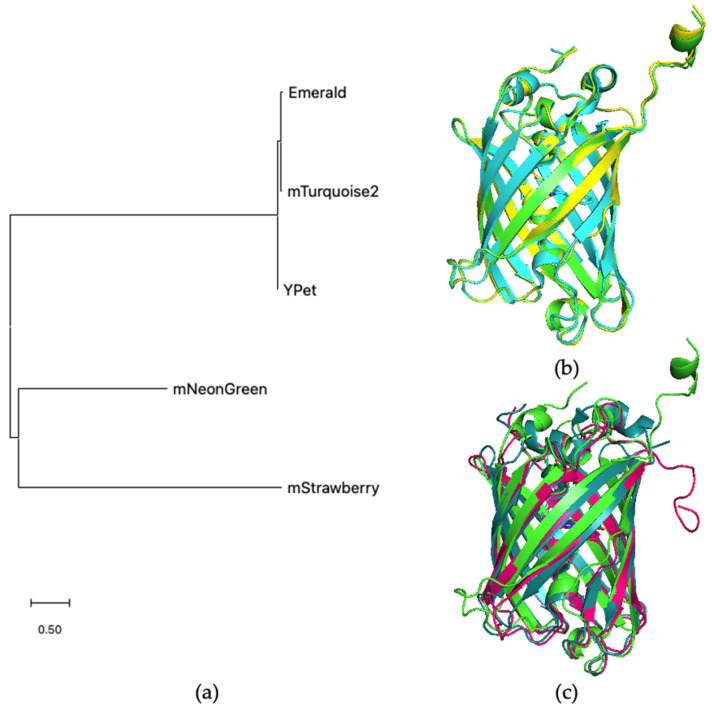
**Structural comparison of FPs.** (**a**) MEGA generated a phylogenetic tree for five FPs: Emerald, mTruquoise2, Ypet, mNeonGreen, and mStrawberry. (**b**) AlphaFold 2.1.0 [36] model of Emerald(green) was compared to mTurquoise2 (cyan) structure (PDB 6YLO) and Alphafold model of Ypet (yellow). (**c**) structure of the Emerald (green) model was also compared to the mNeonGreen (dark green) structure (PDB 5LTR) and the mStrawberry (pink) structure (PDB 2H5P).

**Table 1 molecules-27-05248-t001:** List of FPs selected from FPbase.org (accessed on 18 July 2022).

Fluorescent Protein	λex	λem	Brightness *(**εϕ*)	Oligomerization
AausFP1	504	510	164.9	Dimer
RRvT	556	583	117.9	Tandem dimer
tdTomato	554	581	95.2	Tandem dimer
mNeonGreen	506	517	92.8	Monomer
mClover3	506	518	84.8	Monomer
YPet	517	530	80.1	Weak dimer
mScarlet	569	595	70.0	Monomer
mVenus	515	527	66.6	Monomer
mEos4b	505	516	65.7	Monomer
mKO2	551	565	39.56	Monomer
Emerald	487	509	39.1	Monomer
mOrange2	549	565	34.8	Monomer
mTurquoise2	434	474	27.9	Monomer
mStrawberry	549	565	26.1	Monomer
eGFP	488	507	33.5	Weak dimer
avGFP	395	509	19.8	Dimer
mCherry	574	596	15.8	Monomer

**Table 2 molecules-27-05248-t002:** Microscope filter set component list.

Set	Excitation Filter	Mirror	Emission Filter
1	BrightLine Fluorescence filter 635/18	FF652-DI01	BrightLine Fluorescence filter 680/42
2	RPB550-580 235297	XF2086 580DRLP	BrightLine Fluorescence filter 641/75
3	BrightLine Fluorescence filter 578/21	FF596-DI01	BrightLine Fluorescence filter 641/75
4	BrightLine Fluorescence filter 531/40	FF562-DI03	BrightLine Fluorescence filter 593/40
5	BrightLine Fluorescence filter 509/22	FF526-DI01	BrightLine Fluorescence filter 544/24
6	BrightLine Fluorescence filter 472/30	XF2443	BrightLine Fluorescence filter 525/45
7	BrightLine Fluorescence filter 474/27	XF2443	BrightLine Fluorescence filter 520/35
8	BrightLine Basic Fluorescence filter 434/17	MD-453	BrightLine Basic Fluorescence filter 479/40

## Data Availability

Not applicable.

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
