# Peer review of "21 Fluorescent Protein-Based DNA Staining Dyes"

_molecules, 2022, doi:10.3390/molecules27165248_

Round 1

Reviewer 1 Report

Kim et al. provide a useful characterisation of 21 FPs conjugated to DNA binding proteins for examination of novel fluorescent stains for DNA.

This manuscript is suitable for publication in Molecules.

I have mostly minor concerns which I detail below. The more major one is that I do not find Fig. 5 particularly enlightening to show an alignment and a phylogeny from MEGA as some justification for similar function in the assay in Fig. 4. Elsewhere structural prediction was used, it should be relatively straightforward to make this figure a structural comparison of the dyes to show similarity directly, rather than calculating a phylogeny, to show similarity, for things that were man-made in any case.

My other concern is that there is talk of DNA-specific staining, ie, affinity for a specific sequence. But there is no mention of DNA-PAINT anywhere. Using a dye conjugate to a single strand of DNA (ssDNA) that is complementary, such as in FISH or in DNA-PAINT needs to be mentioned for comparison.

Minor concerns:

L33 - they are easy not it is easy

L43 - I think some limitations of fluorescent proteins would be appropriate here.

L87 - sentence incomplete? specific staining.

L101 - should be C-terminal tDTomato-HMG?

L109 - not clear at this stage what these 'full-stained' images are of?

L111 - should be PDB not DPB.

L128 - lower amount, not less amount

L143 - these guide lines are not quantitative enough. Surely these measurements are correlated and a regression and report of that linear regression should be made. Rather than a 'guide'.

L167 - language in caption not clear, 'The fluorescence intensity measurements between... from microscopy images'. Please clarify.

L181-L190 - I think some mention of DNA PAINT or DNA-based imaging needs to be present here. For example: PMCID: PMC6315775

L216-L225 - as above, I don't find the half page sequence alignment or phylogeny useful. It is also not clear what is meant by 'similarity in the salt concentration at which ... appeared'. Please clarify why they are similar.

L223 -FP sequence deviation would depart how... <- language needs fixing.

L348 - surprised using LED light source not collimated ie laser source, how does that effect fluorophore efficency?

L372 - this is a large number of authors for what seems a straightforward study. What does 'performed research' mean. Can you expand on this acknowledgement? 7 authors involved in plasmid construction seems a lot.

Author Response

Kim et al. provide a useful characterization of 21 FPs conjugated to DNA binding proteins for examination of novel fluorescent stains for DNA.

This manuscript is suitable for publication in Molecules.

>> We appreciated your positive evaluation.

I have mostly minor concerns which I detail below. The more major one is that I do not find Fig. 5 particularly enlightening to show an alignment and a phylogeny from MEGA as some justification for similar function in the assay in Fig. 4. Elsewhere structural prediction was used, it should be relatively straightforward to make this figure a structural comparison of the dyes to show similarity directly, rather than calculating a phylogeny, to show similarity, for things that were man-made in any case.

≫ As suggested, we changed Figure 5 using MEGA and Alphafold 2.1.0.

Figure 5. Structural comparison of FPs. (a) MEGA generated phylogenetic tree for five FPs: Emerald, mTruquoise2, YPet, mNeonGreen, and mStrawberry. (b) AlphaFold 2.1.0 [36] model of Emerald (green) was compared to mTurquoise2 (blue) structure (PDB 6YLO) and Alphafold model of Ypet (yellow). (c) structure of Emerald (green) model was also compared to mNeonGreen structure (PDB 5LTR) and mStrawberry structure (PDB 2H5P).

My other concern is that there is talk of DNA-specific staining, ie, affinity for a specific sequence. But there is no mention of DNA-PAINT anywhere. Using a dye conjugate to a single strand of DNA (ssDNA) that is complementary, such as in FISH or in DNA-PAINT needs to be mentioned for comparison.

>> We newly added FISH and DNA-PAINT as follows.

One of the important advantages of FP-DBP as a dye is its affinity toward a specific DNA sequence. Nonetheless, selective binding exclusively toward a specific sequence on a single molecular level has been a challenging task [37]. In addition to the optical mapping system described in the introduction, fluorescence in-situ hybridization (FISH) has been a tool to obtain sequence information from metaphase chromosomes. Further, DNA points accumulation for imaging in nanoscale topography (DNA-PAINT) can enhance the resolution of DNA images [38, 39]. However, the hybridization method is based on DNA melting to expose single-stranded DNA. In contrast, DNA binding proteins can recognize the sequence without opening the double-helix. Thus, we chose protein to direct specific binding; transcription activator-like effector (TALE) and Zinc-Finger domain (ZnF), for instance, are known for their specific sequence affinity.

Minor concerns:

L33 - they are easy not it is easy

>> We corrected it.

L43 - I think some limitations of fluorescent proteins would be appropriate here.

>> We added the following sentence to mention FP limitation with a recent paper.

As described so far, FPs have many advantages, but they also have disadvantages. A primary limitation of FP is low stability, especially photostability, attributed to proteins’ properties such as denaturation or degradation. Therefore, there have been efforts to solve this problem through genetic engineering. Recently, a highly stable green fluorescent protein, StayGold, was reported to overcome the durability limitation [10].

L87 - sentence incomplete? specific staining.

>> We completed the sentence as follows

Using tTALE-FPs, we also demonstrated that A/T-specific staining is also possible with other FPs.

L101 - should be C-terminal tDTomato-HMG?

>> We added more explanation to clarify the meaning as follows.

However, when HMG was linked at C-terminus (tdTomato-HMG), the expression was successful.

L109 - not clear at this stage what these 'full-stained' images are of?

>> We changed the sentence to clarify the meaning as follows.

FP-DBP stained λ phage DNA images were given below.

L111 - should be PDB not DPB.

>> We used the abbreviation of DNA binding protein as DBP.  

DNA binding protein (DBP) structure models were colored black and were aligned to DNA by using PDB 2EZD

L128 - lower amount, not less amount

>> Corrected as was suggested.

L143 - these guide lines are not quantitative enough. Surely these measurements are correlated and a regression and report of that linear regression should be made. Rather than a 'guide'.

>> We drew new lines based on the linear regression. Also, texts were fixed accordingly.

Figure 2 caption
The dotted lines are calculated for a linear relationship. R2 of the linear regressions was 0.48 for HMG constructs and 0.08 for tTALE constructs.  

In the result section
The dashed line calculated for a linear relationship between FP brightness and DBP-FP fluorescence intensity can be a general guideline. FPs linked to HMG correlated better than FPs linked to tTALE. Because tTALE is larger than HMG, tTALE is more prone to affect the brightness of an FP. And the correlation of tTALE constructs reflected that.

L167 - language in caption not clear, 'The fluorescence intensity measurements between... from microscopy images'. Please clarify.

>> We emphasized the positional effects for the Figure 3 caption.

Figure 3. Positional effects of FP and DBP. (a) Integrated intensity comparison of HMG-RRvT vs. RRvT-HMG. The λ DNA molecules were stained with FP-DBP. (b) Stretching comparison of HMG-mNeonGreen (left) vs. mNeonGreen-HMG (right). Scale bars = 10 μm

L181-L190 - I think some mention of DNA PAINT or DNA-based imaging needs to be present here. For example: PMCID: PMC6315775

>> As mentioned earlier, we added a few sentences.

L216-L225 - as above, I don't find the half page sequence alignment or phylogeny useful. It is also not clear what is meant by 'similarity in the salt concentration at which ... appeared'. Please clarify why they are similar.

>> We revised Figure 5 as mentioned earlier

L223 -FP sequence deviation would depart how... <- language needs fixing.

>> We revised the paragraph including the sentence as follows

To explain the differences in DNA staining patterns of different tTALE-FPs, we compared protein sequences and structures. Figure 5A shows a phylogenetic tree generated by MEGA (molecular evolutionary genetic analysis) software. Emerald, mTurquoise2, and YPet are closely related. Figure 5B shows the overlapped structure of three FPs, which are almost identical to one another. In contrast, mStrawberry and mNeonGreen were significantly apart from the three FPs. Figure 5c shows overlapped structure of Emerald, Strawberry, and mNeonGreen. They show differences among them. These differences may explain the different profiles of DNA-bound tTALE-FPs in Figure 4.

L348 - surprised using LED light source not collimated ie laser source, how does that effect fluorophore efficiency?

>> We have three laser systems (488, 514, 647 nm), but recently, we only used LED light source. The primary reason was that laser systems were too expensive to cover various FPs. Further, it was cumbersome to align each laser beam for a different FP. Our LED light source (SOLA SM II) is so bright and white in color. Thus, for our expansion of different FPs, selecting excitation wavelength through filtering ‘white’ LED light source was more efficient for our set-up than getting different laser sources for different wavelengths. We found LED light source had enough excitation strength for our observation.

L372 - this is a large number of authors for what seems a straightforward study. What does 'performed research' mean. Can you expand on this acknowledgement? 7 authors involved in plasmid construction seems a lot.

>> I understand your concern. However, I wish to explain that the manuscript was possible through the works of multiple students. And I think it is necessary to give credit for their efforts.

Reviewer 2 Report

1.       This manuscript provides a number of FPs that stained DNA molecules, however, little evidence demonstrated whether the binding of these FPs with DNA molecules would cause disturbance to DNAs, gene expression for example. Especially the author demonstrated that N-ter or C-ter placements would lead to different behaviors of DNA on positively charged surface.

2.       The author demonstrated that the positively charged surface was applied in microscope imaging. However, since the FPs emitted bright fluoresce when they are not bound to DNA molecules, it is hard to tell the fluorescence on the surface are derive from FPs attached to DNAs or simply FPs attached to the surface.

3.       The fluorescent features such as fluorescence intensities quantifications of the recombined FP should also be examined with fluorescence spectra, other than testing with microscope and quantified with ImageJ.

4.       DNA sequences utilized in this article should be displayed in this manuscript, for the reason that different DNA own different lengths, different sequences, or even secondary structures, which could have great impact on the binding of FPs with DNAs.
